# Designing a Resilient–Sustainable Supply Chain Network of Age-Differentiated Blood Platelets Using Vertical–Horizontal Transshipment and Grey Wolf Optimizer

**DOI:** 10.3390/ijerph20054078

**Published:** 2023-02-24

**Authors:** Mohammad Shokouhifar, Alireza Goli

**Affiliations:** 1Department of Electrical and Computer Engineering, Shahid Beheshti University, Tehran 1983969411, Iran; 2Department of Industrial Engineering and Future Studies, University of Isfahan, Isfahan 8174673441, Iran

**Keywords:** platelets, resiliency, sustainability, supply chain management, COVID-19, optimization, shortage, wastage, blood transshipment, grey wolf optimizer

## Abstract

Blood platelets are a typical instance of perishable age-differentiated products with a shelf life of five days (on average), which may lead to significant wastage of some collected samples. At the same time, a shortage of platelets may also be observed because of emergency demands and the limited number of donors, especially during disasters such as wars and the COVID-19 pandemic. Therefore, developing an efficient blood platelet supply chain management model is highly necessary to reduce shortage and wastage. In this research, an integrated resilient–sustainable supply chain network of perishable age-differentiated platelets considering vertical and horizontal transshipment is designed. In order to achieve sustainability, economic cost, social cost (shortage), and environmental cost (wastage) are taken into account. A reactive resilient strategy utilizing lateral transshipment between hospitals is adopted to make the blood platelet supply chain powerful against shortage and disruption risks. The presented model is solved using a metaheuristic based on a local search-empowered grey wolf optimizer. The obtained results demonstrate the efficiency of the proposed vertical–horizontal transshipment model in reducing total economic cost, shortage, and wastage by 3.61%, 30.1%, and 18.8%, respectively.

## 1. Introduction

Blood plays the role of fuel for the survival of patients and for saving lives [1]. Generally, four components are extracted from blood: fresh frozen plasma (FFP), white blood cells, red blood cells (RBC), and platelets [2]. Each of these components has a specific shelf life. Based on the preservation solution, the average shelf life of FFP is one year, the shelf life of RBC is 40 days, and the shelf life of blood platelets is only five days on average [3]. Platelets have the shortest shelf life and are widely used in various medical treatments, such as bone marrow failure, cancer treatment, cardiovascular surgery, organ transplantation, traumatology, hematopoietic stem cell transplantation, AIDS, and Hepatitis [4]. Platelets are segregated from other components of blood using buffy coats, platelet-rich plasma, or apheresis [5]. 

Platelets are a typical instance of perishable age-differentiated products [6]. Based on the type of blood platelet demands, they can be categorized into critical illness patients and less critical patients [7]. For treating critical patients with acute blood loss and those with lung, liver, kidney, and heart dysfunction, fresh (young) platelets are highly preferred by physicians, as they can significantly enhance their platelet count and prolong the time intervals between transfusions [8]. However, in the treatment of less critical cases (general surgeries and traumatology), any platelet age (young, mature, or old) will serve the purpose [6].

Inventory management of perishable products, such as food crops, vegetables, fruits, kinds of seafood, and medicines, is a challenging problem due to the specific characteristics of these products [9]. This issue becomes more complex when such products encounter quality deterioration during their shelf life, e.g., blood platelets [10]. Hospitals may face a shortage of blood platelets, which can postpone or cancel major medical operations. On the other hand, excessive storage of blood platelets leads to significant wastage due to expiration, which incurs enormous economic and environmental costs for the health system [11]. Due to the short shelf life of platelets and limited human-based supply resources (i.e., donors), the design of an efficient blood platelet supply chain network (BPSCN) is highly required [12]. In this regard, efficient mechanisms must be used to optimize the inventory level of such perishable products and avoid their shortage and wastage (more sustainability) [13].

Another challenge in designing BPSCNs is associated with disruptions and disasters, such as natural disasters, operational risks, political events, wars, and pandemics, which significantly affect the operation of health systems and cause irreparable damage [14]. Thus, a primary factor in designing BPSCNs is accurately identifying these disruption risks and finding proper resilient solutions to cope with these risks [15]. Generally, sustainability and resiliency are two main concepts that must be simultaneously taken into account to ensure the system’s survival. To maintain the supply chain sustainability performance, it should be resilient against disruption risks [16].

In this study, an integrated resilient–sustainable BPSCN of age-differentiated (young, mature, and old) platelets considering vertical–horizontal transshipments is presented. To achieve this purpose, a multi-objective, multi-period, and multi-age mathematical model considering sustainability aspects covering the total economic cost and social and environmental issues is formulated. The vertical transshipment from blood banks to hospitals is used for planning demands, while the horizontal transshipment between different hospitals is applied for real-time emergency demands (in the case of shortage). To enhance the resiliency of the BPSCN model under the conditions of disruption, a reactive resilient strategy utilizing lateral transshipment is developed. Moreover, this policy can improve inventory management by simultaneously reducing shortage and waste, consequently minimizing operation costs and enhancing sustainability. Finally, a local search empowered grey wolf optimizer utilizing global and local search mechanisms is proposed as a metaheuristic-based solution method to optimize the established model. 

The rest of the paper is organized as follows. The related literature is reviewed in Section 2. The proposed model for the BPSCN design is formulated in Section 3. Our solution method based on the LSGWO algorithm is proposed in Section 4. Obtained results by the proposed method and comparison with the existing techniques are reported in Section 5. Finally, this paper will conclude with some future works in Section 6. 

## 2. Literature Review

A BPSCN involves the collection of blood samples from donors, testing at blood banks, and delivering them to hospitals. The functions of a blood bank include processing and testing the received blood samples and splitting them into various components, storing and/or issuing them to hospitals on the basis of requirements, and disposal of expired samples [17]. The collected samples should pass various tests, such as HIV, HBV, HCV, Malarial Parasite, West Nile Virus, and NAT test, before undergoing transfusion to the hospitals, thereby ensuring their quality and safety for patients [18]. In some cases, e.g., during the COVID-19 pandemic, some additional tests, such as antibody tests, may also be requested. The abovementioned processes take about one or two days, and valuable components are made available for patient use in hospitals [19]. All these activities are time-consuming and involve considerable costs. Therefore, the collected blood has to be managed judiciously, as there is no alternative for blood. These factors, along with the concise shelf life of platelets and uncertainties in supply/demand, make the management of BPSCNs a complex task. The specific physiological characteristics of blood include:*Scarcity of blood*: The human body is the sole supply channel of blood. The required blood in hospitals should be extracted from voluntary donors and cannot be replaced by other materials.*Shortage*: Blood products are critical to ensure health and the saving of lives. Blood shortage may lead to serious consequences, especially in the case of critical illness patients.*Wastage*: Blood products, especially platelets, are highly perishable. If platelets spoil because of infection or expiration, they must be disposed by stipulated protocols. Excessive discarding of infected and outdated platelets leads to extra costs and negative environmental impacts.*Age-differentiated demands*: Some treatments need high blood platelet freshness, while less critical patients can be treated via older platelets.

### 2.1. Inventory Management Methods

Inventory management of blood platelets involves additional complexities due to the limited shelf life [20]. In a routine delivery, processes of blood collection and transfusion are carried out via bottom-up information (lower: hospitals; upper: blood bank). In this regard, hospitals or other demand nodes usually send their requests to the blood bank at the end of each day, according to their inventory level and demands. Then, the blood bank periodically schedules the received planned orders and delivers blood products to the hospitals early in the morning [1]. This periodicity is mostly considered as one day [19]. In some studies [21,22,23], a fixed lead time (from sending order to the blood bank to receiving products in the hospital) of one day is considered. In addition to the planned ordering in routine delivery, emergency or unscheduled orders may also be requested by hospitals to replenish their inventory in the case of shortage [24].

Most existing models studied blood inventory using periodical policies from the perspective of perishable products. Baron et al. [25] proposed a heuristic based on (S, s) policy by continuous inventory inspection, where S and s represent the target inventory level and reorder point, respectively. Broekmeulen et al. [26] developed a (R, s, nQ) policy considering the shelf life of products, in which a replenishment order of n packs each with size Q is requested when the inventory level is below the reorder level s. Civelek et al. [6] categorized demands into young, mature, and old according to the residual shelf life of platelets. They allowed the replacement of each type of demand with each other considering an additional cost. Dillon et al. [27] developed a periodic inventory replenishment policy for platelets, while considering perishability, lead time, and multiple types of platelets. Shokouhifar et al. [2] proposed a (R, Q) policy in which an order with quantity Q is sent to the blood bank when the inventory level drops below the reorder level R. They considered fuzzy supply/demand uncertainties and applied the Whale Optimization Algorithm (WOA) to tune the inventory management model. 

Issuing policy of blood platelets has a significant effect on wastage and shortage of platelets [2]. Last-in–first-out (LIFO) [28] and first-in–first-out (FIFO) [24] are two popular issuing policies used in BPSCNs. To reduce wastage rate, the oldest inventory must be issued first (i.e., FIFO), while the most recently stored (fresher) platelets are preferred by most physicians, especially for critical cases (i.e., LIFO). In order to gain the advantages of both methods, some hybrid FIFO–LIFO techniques were also proposed [29,30] by splitting the inventory into two stocks, each utilizing a different issuing policy.

### 2.2. Supply Chain Management Methods 

Sustainable design of supply chains can help organizations achieve competitive advantages [31,32]. Gunpinar and Centeno [33] proposed a two-stage BPSCN comprising a hospital and a blood bank, and presented an integer programming model considering platelet freshness. Rajendran and Ravindran [20] utilized a genetic algorithm to optimize a two-echelon stochastic integer programming model for the sustainable BPSCN design considering demand uncertainty and FIFO issuing policy. Mousavi et al. [34] proposed a stochastic programming model to design a BPSCN where social and environmental issues were taken into account. Abbaspour et al. [5] used a Markov model to optimize a BPSCN under scheduled and emergency demands and a hybrid FIFO–LIFO issuing policy, while freshness and expiration of platelets were also under consideration. 

Robust optimization has been used by several researchers to cope with uncertainties occurring in BPSCNs. Hamdan and Diabat [35] used a robust multi-objective optimization to minimize the time and cost of blood delivery from blood bank to hospitals in BPSCNs. Sohrabi et al. [36] proposed a multi-objective model under elective and non-elective platelet demands, substitution allowance, medical urgency, and the age of platelets. They utilized a robust fuzzy stochastic model and applied a combined metaheuristic based on a genetic algorithm and simulated annealing to solve the model.

The resilient design of supply chains is of utmost importance to make the supply chain powerful against disruption risks [37]. Wang and Chen [38] developed a robust optimization model for a resilient BPSCN to handle uncertain demands under disaster conditions. Ivanov [39] proposed a simulation-based method to analyze the role of sustainability and environmental issues in designing resilient supply chains under disruption. Pavlov et al. [40] presented a resilient–sustainable supply chain network using a network redundancy model and a proactive resilient strategy. Elluru et al. [41] developed a location-routing model considering various proactive and reactive strategies to design a resilient supply chain network. Ash et al. [42] proposed a robust multi-objective model to enhance the resiliency of supply chains during the COVID-19 pandemic.

Moreover, resiliency in BPSCNs has been noticed by some researchers. Khalilpourazari et al. [43] considered the transportation planning problem in a BPSCN under disruptions caused by natural disasters such as earthquakes. They used different modes, including helicopters, for collecting and transporting blood, aiming to reduce the shortage. Tirkolaee et al. [44] proposed a bi-objective socio-economic model to minimize economic costs and maximize job opportunities in designing a resilient BPSCN, taking into account the adverse effects of COVID-19. Shokouhifar and Ranjbarimesan [14] proposed a resilient BPSCN with the help of blood supply/demand forecasting using a multivariate time-series analysis considering new confirmed cases/deaths during COVID-19.

### 2.3. Transshipment Methods

By considering blood transshipping factors, different transshipments can be employed as vertical (from blood banks to hospitals) or horizontal (among different blood banks or different hospitals) [1]. In a normal case, vertical transshipment is made up of planned orders of hospitals. In the case of shortage, horizontal transshipment effectively relieves blood shortage, affecting inventory decision-making in blood banks and hospitals. Horizontal blood transshipment among blood banks can be used to compensate for the lack of local collection and balance inventory of blood banks [45].

Various transshipment techniques have been presented for regular and perishable products, but there are few pieces of research on blood products. Wang and Ma [46] established an optimization method for solving emergency transshipping of blood products, which aims to maximize the freshness of delivered items. Wang and Ma [7] proposed an emergency blood transshipment mechanism between different blood banks, considering different ages of the products. Puranam et al. [47] presented a hospital network model utilizing transshipment between hospitals which a joint blood bank supports. Shokouhifar et al. [2] proposed another horizontal transshipment between hospitals to satisfy emergency and unscheduled demands, demonstrating that hospital collaboration not only avoids shortage, but also reduces the wastage rate.

### 2.4. Research Gaps and Our Contributions

Despite different pieces of research over the past years, there are still several gaps in the resilient and sustainable design of BPSCNs. Regarding previous works conducted in BPSCN design, it is observed that few papers considered blood platelets as age-differentiated perishable products. In the majority of existing methods, substituting old platelets for young platelets is allowed. To the best of the authors’ knowledge, there is no horizontal transshipment between hospitals that has been used in BPSCNs for resiliency. Based on the publications in recent years, sustainability and resiliency are a heated debate among researchers. By outbreaking the COVID-19 pandemic, resiliency and sustainability are of utmost importance in the design of supply chain networks [48]. However, most of the existing methods in BPSCNs consider these two concepts separately.

To cover the existing gaps in the literature, we present a reactive resilient strategy utilizing lateral transshipment between hospitals in designing a resilient–sustainable BPSCN of age-differentiated platelets covering both planned and emergency demands. The proposed model improves the performance of the BPSCN in terms of economic, social, and environmental issues by making a trade-off among different objectives using a multi-objective weighted average method. The aim is to assist the decision maker in achieving more resilient and sustainable processes and reducing shortage and wastage. From the methodological point of view, we propose a local search empowered GWO (LSGWO) is presented to gain the global search ability of GWO and multiple local searches into the integrated LSGWO algorithm. Our motivation is to achieve higher performance in terms of objective functions with faster convergence.

## 3. Problem Statement

### 3.1. Blood Platelet Supply Chain Network Design

In this research, the BPSCN design of age-differentiated platelets in a three-echelon network is assessed, considering vertical–horizontal transshipments, comprising D donor sites, M blood banks, and N hospitals. The problem is discussed in T days. The studded supply chain network and the flows among the elements of the network are demonstrated in Figure 1. Each blood bank is supplied by some donor sites, while a donor site is connected to a single blood bank. In a lower stage, each hospital is connected to a single blood bank, but a blood bank can support several hospitals. The donor site d delivers $S_{dt}$ products at time period (day) t to the blood bank j. The blood bank *j* processes and tests the received blood samples and extracts platelets. Next, it delivers only $q_{j}\%$ of the useful platelets to hospitals after a Testing-Producing Time $TPT_{j}$. The quality inspection of platelets by the blood bank should start after the collection of raw materials, and this process should not take more than two days to reach the hospital. For the aim of simplicity, it is considered as a fixed value of two days in this study. The hospital i faces a planned (scheduled) demand $PD_{ilt}$ and an emergency (unscheduled) demand $ED_{ilt}$ for platelet age l at the day t. Based on different reasons, e.g., disasters, disruptions, and emergency demand, supply and/or demand in BPSCNs should be considered as uncertain parameters. To deal with these uncertainties, we present a reactive resilient strategy using lateral transshipment between hospitals, which satisfies the real-time shortages from other hospitals. To formulate this problem, the following conditions are assumed: Each donor site has a limited capacity for supplying platelets to the blood bank.On average, $q_{j}\%$ of the received samples in each blood bank *j* are useful.Every day, each hospital faces two types of demands for each platelet age: planned (scheduled) demand and emergency (unscheduled) demand.Each hospital is supplied by a specific blood bank.Each blood bank may support one or more hospitals.In the case of shortage for emergency orders and/or disruptions, each hospital can interchange with other hospitals for any platelet age.The received blood samples are processed and tested in blood banks in two days.The received platelets in hospitals are considered with a shelf life of three days.Platelets and demands are grouped into three categories: young, mature, and old.The shelf life of fresh (young) platelets received by a hospital is three days.Young products become mature in one day.Mature products become old in one day.Old platelets are expired in one day.The issuing policy is considered as First Input First Output (FIFO) in both blood banks and hospitals.Hospitals request young platelets from the associated blood banks.Hospitals can request any ages of platelets from other hospitals.Lead time of transportation from donor sites to blood banks is zero.Lead time of transportation from blood banks to hospitals is one day, i.e., the blood bank delivers the received orders to hospitals early in the morning of the next day.Lead time of horizontal exchange between hospitals is neglected.

### 3.2. Notations

Indices, parameters, and decision variables of the proposed BPSCN model can be summarized in Table 1. Decision variables include blood bank selection for each hospital $X_{ij}$, lateral transshipment network design $X_{ik}^{LT}$, allocation of platelet sources of hospitals from blood banks $Y_{ijlt}$ and other hospitals (i.e., lateral transshipment) $Y_{iklt}^{LT}$, reorder point $R_{ijl}$ and order quantity $Q_{ijl}$ of hospitals for different platelet ages from blood banks, and similarly, $R_{ikl}^{LT}$ and $Q_{ikl}^{LT}$ for ordering from other hospitals. It is assumed that each hospital follows a periodic (everyday) review process using (R,Q) policy.

### 3.3. Reactive Resilient Model

As the lead time of planned orders from the blood bank to the hospital is one day, a shortage of platelets for unscheduled emergency demands may be observed. In addition to the emergency demands, disruption risks may result in hospital shortages. In this paper, we study two types of disruptions, i.e., breakdown in blood banks and transportation links between blood banks and hospitals. Both lead to the impossibility of transferring platelets from the associated blood bank to the hospital. More specifically, other blood banks could not be called to satisfy the real-time demands due to their lead time of one day. To make the BPSCN resilient against the disruption sources and emergency demands, we introduce a lateral transshipment between hospitals with a lead time of zero to satisfy real-time shortages in a hospital with the help of other hospitals. 

In a normal case, the required platelets for a hospital are supplied via planned orders from the associated blood bank. However, in the case of any shortage or disruptions in providing platelets by the blood bank caused by breakdowns in the blood bank or in the transportation link, a reactive resilient strategy is performed to compensate for the boosted shortage by emergency demands and/or disruptions. In the proposed method, the shortage of platelets in hospital i for the platelets with age *l* on day *t* can be sourced through real-time ordering from hospital *k* on the same day *t*. As the lead time of lateral transshipment is considered zero, all requests would be delivered on the same day of ordering. Of course, it boosts extra costs of ordering and transportation between hospitals, but it can be helpful to minimize the shortage of the target hospital and avoid wastage of the hospitals with high inventory. In other words, the proposed resilient strategy leads to achieving a better shortage–wastage balance by simultaneously minimizing the shortage and wastage of hospitals. 

### 3.4. Inventory Management Model

Every day, each blood bank collects all blood samples from its associated donor sites. After processing and testing the received samples, platelets are extracted at a young age and would be ready to be delivered to hospitals after $TPT_{j}$ days. The inventory of blood bank j for young platelets on day t can be formulated as a summation of useful platelets as Equation (1).
(1)$$IB_{jlt}=\overset{D}{\underset{d=1}{\sum }}q_{j}S_{dt-TPT_{j}}Z_{jd}\;\;\;\;l=1$$

The inventory of platelets with mature or old ages is also formulated as inventory of platelets with fresher age (l−1) from yesterday (t−1) minus delivered platelets with age (l−1) yesterday (t−1) from the blood bank *j* to all associated hospitals on day t, which can be calculated as Equation (2).
(2)$$IB_{jlt}=IB_{jl-1t-1}-\overset{N}{\underset{i=1}{\sum }}Q_{ijl}Y_{ijl-1t-1}\;\;\;\;\forall l>1$$

In a hospital, the inventory of young platelets can be calculated as the total received fresh platelets from its associated blood bank according to Equation (3). In contrast, the inventory of other platelet ages is formulated as the summation of the transferred platelets of fresher platelets from yesterday and the total received platelets of the same age on the same day from the associated blood bank according to Equation (4).
(3)$$I_{ilt}^{0}=\overset{M}{\underset{j=1}{\sum }}Q_{ijl}Y_{ijlt}+\overset{N}{\underset{k=1}{\sum }}Q_{ikl}^{LT}Y_{iklt}^{LT}-\overset{N}{\underset{k=1}{\sum }}Q_{kil}^{LT}Y_{kilt}^{LT}\;\;\;\;\forall l=1$$
(4)$$I_{ilt}^{0}=I_{il-1t-1}+\overset{M}{\underset{j=1}{\sum }}Q_{ijl}Y_{ijlt}+\overset{N}{\underset{k=1}{\sum }}Q_{ikl}^{LT}Y_{iklt}^{LT}-\overset{N}{\underset{k=1}{\sum }}Q_{kil}^{LT}Y_{kilt}^{LT}\;\;\;\;\forall l>1$$

Moreover, the inventory of platelets with age *l* at the end of day t in hospital i (after satisfying both planned and emergency demands) can be formulated as Equation (5).
(5)$$I_{ilt}=\left(I_{ilt}^{0}-D_{ilt}\right)\left(1-E_{ilt}\right)$$

### 3.5. Objective Function

The proposed model aims to consider extra costs incurred by the age-differentiated platelets and lateral transshipment in addition to the workflow costs. The proposed three-echelon BPSCN model is formulated using a mixed integer linear programming and a multi-objective function to minimize economic cost while considering social (shortage) and environmental (wastage) objectives to be minimized.

#### 3.5.1. Economic Objective Function

Economic objective function $F_{EC}$ comprises four costs, including the cost of ordering from blood banks via vertical transshipment ($F_{ec1}$), cost of lateral transshipment between the different hospitals ($F_{ec2}$), cost of transportation ($F_{ec3}$), and cost of inventory holding ($F_{ec4}$), which can be formulated as:(6)$$F_{EC}=F_{ec1}+F_{ec2}+F_{ec3}+F_{ec4}$$


*Cost of ordering from blood banks:*


Every day, hospital i may order a quantity $Q_{ijl}$ of platelets age l from the associated blood bank j. This boasts a fixed and variable ordering cost, as shown in Equation (7).
(7)$$F_{ec1}=\overset{T}{\underset{t=1}{\sum }}\overset{N}{\underset{i=1}{\sum }}\overset{M}{\underset{j=1}{\sum }}\left(f_{j}+\overset{L}{\underset{l=1}{\sum }}c_{jl}Q_{ijl}\right)Y_{ijlt}$$


*Cost of lateral transshipments:*


Due to emergency cases and/or disruptions, the demand of some hospitals may be more than the planned order. On the other hand, some hospitals may face wastage of platelets due to the cancellation of some treatments. The interchange of platelets between hospitals is applied to achieve a better balance between hospitals and make the BPSCN resilient. The cost of lateral transshipment can be expressed as Equation (8).
(8)$$F_{ec2}=\overset{T}{\underset{t=1}{\sum }}\overset{N}{\underset{i=1}{\sum }}\overset{N}{\underset{k=1}{\sum }}\overset{L}{\underset{l=1}{\sum }}c_{kl}^{LT}Q_{ikl}^{LT}Y_{iklt}^{LT}$$


*Cost of transportation:*


Transportation cost comprises fixed and variable costs, including costs of carrying platelets from blood banks to the hospitals ($CT_{B,H}$) and between hospitals ($CT_{H,H}$), which can be formulated as $F_{ec3}=CT_{B,H}+CT_{H,H}$ in Equation (9).
(9)$$F_{ec3}=\overset{T}{\underset{t=1}{\sum }}\overset{N}{\underset{i=1}{\sum }}\overset{M}{\underset{j=1}{\sum }}\overset{L}{\underset{l=1}{\sum }}\left(p_{ij}+r_{ij}d_{ij}\right)Y_{ijlt}+\overset{T}{\underset{t=1}{\sum }}\overset{N}{\underset{i=1}{\sum }}\overset{N}{\underset{k=1}{\sum }}\overset{L}{\underset{l=1}{\sum }}\left(p_{ik}^{LT}+r_{ik}^{LT}d_{ik}^{LT}\right)Y_{iklt}^{LT}$$


*Cost of inventory holding:*


According to Equation (5), inventory of platelets with age l at the end of day *t* in hospital i is $I_{ilt}$. Therefore, total inventory holding cost can be expressed as Equation (10). It should be noted that the platelets with age *L* (old platelets) are not taken into account in the holding cost, as they are expired and would be wasted the next day.
(10)$$F_{ec4}=\overset{T}{\underset{t=1}{\sum }}\overset{N}{\underset{i=1}{\sum }}\overset{L-1}{\underset{l=1}{\sum }}h_{il}I_{ilt}$$

#### 3.5.2. Social Objective Function

If the total demand by hospital *i* is less than its inventory on day *t*, the demand is fully satisfied. However, in the case of shortage, its demand is partially fulfilled. In this case, the cost of shortage for unsatisfied demand can be expressed as Equation (11).
(11)$$F_{SC}=\overset{T}{\underset{t=1}{\sum }}\overset{N}{\underset{i=1}{\sum }}\overset{L}{\underset{l=1}{\sum }}s_{il}\left(D_{ilt}-I_{ilt}^{0}\right)E_{ilt}$$

#### 3.5.3. Environmental Objective Function

Wastage cost includes the expiration of platelets with a remaining life of zero in both blood banks and hospitals. These platelets should be disposed of following the stipulated protocols. More specifically, the expired platelets with age *L* (old platelets) are wasted. The waste cost at blood banks and hospitals can be formulated as Equation (12).
(12)$$F_{EN}=\overset{T}{\underset{t=1}{\sum }}\overset{M}{\underset{j=1}{\sum }}w_{j}\left(IB_{jLt}-\overset{N}{\underset{i=1}{\sum }}Q_{ijL}Y_{ijLt}\right)+\overset{T}{\underset{t=1}{\sum }}\overset{N}{\underset{i=1}{\sum }}w_{i}I_{iLt}$$

#### 3.5.4. Integration of Objective Functions

The multi-objective function can be formulated by aggregating economic, social, and environmental objectives with a weighted sum approach, which is shown in Equation (13).
(13)$$\begin{array}{cc} \mathrm{minimize}:\;OBJ & =w_{EC}F_{EC}+w_{SC}F_{SC}+w_{EN}F_{EN}\\ & =w_{EC}\\ & \times (\sum_{t=1}^{T}\sum_{i=1}^{N}\sum_{j=1}^{M}\left(f_{j}+\sum_{l=1}^{L}c_{jl}Q_{ijl}\right)Y_{ijlt}+\sum_{t=1}^{T}\sum_{i=1}^{N}\sum_{k=1}^{N}\sum_{l=1}^{L}c_{kl}^{LT}Q_{ikl}^{LT}Y_{iklt}^{LT}+\sum_{t=1}^{T}\sum_{i=1}^{N}\sum_{j=1}^{M}\sum_{l=1}^{L}\left(p_{ij}+r_{ij}d_{ij}\right)Y_{ijlt}\\ & +\sum_{t=1}^{T}\sum_{i=1}^{N}\sum_{k=1}^{N}\sum_{l=1}^{L}\left(p_{ik}^{LT}+r_{ik}^{LT}d_{ik}^{LT}\right)Y_{iklt}^{LT}+\sum_{t=1}^{T}\sum_{i=1}^{N}\sum_{l=1}^{L-1}h_{il}I_{ilt})+w_{SC}\times \left(\sum_{t=1}^{T}\sum_{i=1}^{N}\sum_{l=1}^{L}s_{il}\left(D_{ilt}-I_{ilt}^{0}\right)E_{ilt}\right)\\ & +w_{EN}\times \left(\sum_{t=1}^{T}\sum_{j=1}^{M}w_{j}\left(IB_{jLt}-\sum_{i=1}^{N}Q_{ijL}Y_{ijLt}\right)+\sum_{t=1}^{T}\sum_{i=1}^{N}w_{i}I_{iLt}\right) \end{array}$$

In Equation (13), $w_{EC}$, $w_{SC}$, and $w_{EN}$ are three constant weights ($w_{EC}+w_{SC}+w_{EN}=1$) that specify the relative impacts of the economic, social, and environmental issues within the objective function.

#### 3.5.5. Mathematical Model Constraints

In addition to the objective functions, several constraints follow the assumptions and show the relation between different decision variables. The mathematical model constraints are provided in Equations (14)–(28).
(14)$$\overset{M}{\underset{j=1}{\sum }}Z_{jd}=1\;\;\;\;\forall d\in \left\{1,2,\ldots ,D\right\}$$
(15)$$\overset{M}{\underset{j=1}{\sum }}X_{ij}=1\;\;\;\;\forall i\in \left\{1,2,\ldots ,N\right\}$$
(16)$$X_{ii}^{LT}=0\;\;\;\;\forall i\in \left\{1,2,\ldots ,N\right\}$$
(17)$$\overset{N}{\underset{k=1}{\sum }}X_{ik}^{LT}\geq 0\;\;\;\;\forall i\in \left\{1,2,\ldots ,N\right\}$$
(18)$$\overset{N}{\underset{k=1}{\sum }}X_{ik}^{LT}\leq N-1\;\;\;\;\forall i\in \left\{1,2,\ldots ,N\right\}$$
(19)$$0\leq \overset{M}{\underset{j=1}{\sum }}Y_{ijlt}\leq 1\;\;\;\;\forall i\in \left\{1,2,\ldots ,N\right\},\;l\in \left\{1,2,\ldots ,L\right\},\;t\in \left\{1,2,\ldots ,T\right\}$$
(20)$$0\leq \overset{N}{\underset{k=1}{\sum }}Y_{ijlt}^{LT}\leq 1\;\;\;\;\forall i\in \left\{1,2,\ldots ,N\right\},\;l\in \left\{1,2,\ldots ,L\right\},t\in \left\{1,2,\ldots ,T\right\},i\neq k$$
(21)$$R_{LB}\leq R_{ijl}\leq R_{UB}\;\;\;\;\forall i\in \left\{1,2,\ldots ,N\right\},j\in \left\{1,2,\ldots ,M\right\},l\in \left\{1,2,\ldots ,L\right\}$$
(22)$$Q_{LB}\leq Q_{ijl}\leq Q_{UB}\;\;\;\;\forall i\in \left\{1,2,\ldots ,N\right\},j\in \left\{1,2,\ldots ,M\right\},l\in \left\{1,2,\ldots ,L\right\}$$
(23)$$R_{LB}^{LT}\leq R_{ikl}^{LT}\leq R_{UB}^{LT}\;\;\;\;\forall i,k\in \left\{1,2,\ldots ,N\right\},l\in \left\{1,2,\ldots ,L\right\},i\neq k$$
(24)$$Q_{LB}^{LT}\leq Q_{ikl}^{LT}\leq Q_{UB}^{LT}\;\;\;\;\forall i,k\in \left\{1,2,\ldots ,N\right\},l\in \left\{1,2,\ldots ,L\right\},i\neq k$$
(25)$$IB_{jlt}\geq \overset{N}{\underset{i=1}{\sum }}Q_{ijl}Y_{ijlt}\;\;\;\;\forall j\in \left\{1,2,\ldots ,M\right\},l\in \left\{1,2,\ldots ,L\right\},t\in \left\{1,2,\ldots ,T\right\}$$
(26)$$\overset{M}{\underset{j=1}{\sum }}\overset{D}{\underset{d=1}{\sum }}q_{j}S_{dt-TPT_{i}}Z_{jd}\geq \overset{N}{\underset{i=1}{\sum }}D_{ilt}\;\;\;\;\forall t\in \left\{1,2,\ldots ,T\right\},l=1$$
(27)$$\overset{M}{\underset{j=1}{\sum }}\overset{D}{\underset{d=1}{\sum }}q_{j}S_{dt-TPT_{i}-\left(l-1\right)}Z_{jd}-\overset{N}{\underset{i=1}{\sum }}D_{il-1t-1}\geq \overset{N}{\underset{i=1}{\sum }}D_{ilt}\;\;\;\forall t\in \left\{1,2,\ldots ,T\right\},l=2$$
(28)$$\overset{M}{\underset{j=1}{\sum }}\overset{D}{\underset{d=1}{\sum }}q_{j}S_{dt-TPT_{i}-\left(l-2\right)}Z_{jd}-\overset{N}{\underset{i=1}{\sum }}\left(D_{il-1t-1}-D_{il-2t-2}\right)\geq \overset{N}{\underset{i=1}{\sum }}D_{ilt}\;\;\forall t\in \left\{1,2,\ldots ,T\right\},l=2$$

Constraint (14) expresses that each donor site delivers its collected blood donations to a single blood bank. Constraint (15) ensures that each hospital is sourced by a specific blood bank. Constraints (16)–(18) denote that each hospital may be sourced by one, two, three,…, or N−1 other hospitals through lateral transshipment. Constraint (19) expresses that every day, each hospital may receive the platelets of any age from the blood bank, or not. In the case of shortage of a hospital for any platelet age on each day, constraint (20) ensures that each hospital can be sourced by other hospitals. Constraints (21)–(24) state boundaries for the reorder point and order quantity for purchasing from the blood bank and through lateral transshipment. Constraints (25)–(28) express that there is enough supply to meet the demands of different platelet ages.

## 4. Solution Method

It has been proved that supply chain management models are NP-hard [2,36,44], and thus, metaheuristics can be the best choice to solve this problem. To efficiently solve the proposed BPSCN model and obtain the best trade-off between convergence speed and performance, a local search-empowered grey wolf optimizer (LSGWO) is proposed to gain the advantages of both population-based metaheuristics and local searching into the integrated algorithm. 

In the LSGWO algorithm, GWO is started by an initial population of grey wolves, and then, iteratively, three steps of evaluation of objective function, updating population, and local searching are performed until the stop criterion is achieved, i.e., arriving at the maximum specified number of iterations. The local search improvement phase in the LSGWO algorithm is designed in such a way that the changes in the solution are gradually decreased during the execution of the LSGWO algorithm. Through this method, the better the algorithm performs, the less shaking is applied by the local search operators. The flowchart of the proposed LSGWO algorithm is shown in Figure 2. To gain more insights into the details of the LSGWO algorithm, its pseudo code is provided in Algorithm 1.
**Algorithm 1.** Local Search empowered Grey Wolf Optimizer (LSGWO)**Inputs:**     Number of donor sites (*D*), blood banks (*M*), hospitals (*N*), and time periods (*T*)       Controllable parameters of LSGWO: PopSize, MaxIter, and local search operators**Output:**         Optimized BPSCN model: *X_ij_*, X*^LT^_ik_*, *Y_ijlt_*, *Y^LT^_iklt_*, *R_ijl_*, *Q_ijl_*, *R^LT^_ikl_*, and *Q^LT^_ikl_***LSGWO algorithm:**1.     Initialize a population of grey wolves (feasible solutions)2.     Evaluate *OBJP* for each grey wolf using Equation (29)3.     Update X*_α_* (best solution), X*_β_* (second best solution), and X*_δ_* (third best solution)4.     *it* = 0; 5.     ***while*** (*it* ≤ MaxIter)6.         Update *a*7.         ***for*** each grey wolf8.               Update *A*, *C*, *r*_1_, and *r*_2_ for alpha, beta, and delta9.               ***if*** (|*A*| < 1)10.                     Update the position of grey wolf *g* by *attacking prey* using Equation (30)11.             ***else if*** (|*A*| > 1)12.                     Update the position of grey wolf *g* by *search for prey* using Equation (30)13.             ***end if***14.         ***end for***15.         Evaluate *OBJP* for each grey wolf using Equation (29)16.         Update X*_α_*, X*_β_*, and X*_δ_*, and global best solution X*17.         Perform local search improvement on X* to generate X*_new_18.         Replace X* with X*_new_, if *OBJP**_new_ < *OBJP**19.         *it* = *it* + 1;20.     ***end while*****Return X* as the best solution of the BPSCN model**

### 4.1. Solution Representation (Encoding and Decoding)

As mentioned in Section 3, decision variables include $X_{ij}$, $X_{ik}^{LT}$, $Y_{ijlt}$, $Y_{iklt}^{LT}$, $R_{ijl}$, $Q_{ijl}$, $R_{ikl}^{LT}$, and $Q_{ikl}^{LT}$. A feasible solution to the BPSCN problem is encoded as a hybrid binary–integer structure. Choosing an optimal connection network between hospitals and blood banks to select a proper blood bank for each hospital is represented as a binary matrix of dimension *N* × *M* (i.e., S.$X_{ij}$). Construction of the lateral transshipment network between hospitals is a hybrid assignment problem of dimension *N* × *N* (i.e., S.$X_{ik}^{LT}$). The reorder point and order quantity of each hospital when ordered from the blood bank are defined as two matrices of dimension *N* × *L* (i.e., S.$R_{ijl}$ and S.$Q_{ijl}$). Moreover, reorder point and order quantity of hospitals from other hospitals can be determined as two matrices of dimension *N* × *L* (i.e., S.$R_{ikl}^{LT}$ and S.$Q_{ikl}^{LT}$). The decision variables $Y_{ijlt}$ and $Y_{iklt}^{LT}$ are not encoded into the feasible solutions as they can be derived for each day according to other decision variables and (R,Q) policy. An example of the solution encoding can be seen in Figure 3. 

### 4.2. Initial Population Generation

GWO is a swarm intelligence algorithm which was firstly proposed by Mirjalili et al. [49]. The social behavior of grey wolves in hunting inspired it. As seen in Figure 2, the search process begins by generating a random population containing PopSize solutions. In each iteration, the entire population is justified via an objective function, and then, iteratively, the current population is updated based on the encircling prey strategy utilizing the search for prey (exploration) and attacking prey (exploitation).

### 4.3. Evaluation of Objective Function

To validate each grey wolf, the total objective function (*OBJ*) is calculated by Equation (13). If all constraints within Equations (14)–(28) are fulfilled, the value of the objective function with the penalty (*OBJP*) is equal to *OBJ*. Otherwise, a penalty function is multiplied by the total objective function as Equation (29).
(29)$$OBJP=OBJ\times \left(1+PF\right)$$
where PF can be calculated as the number of unsatisfied constraints.

### 4.4. Population Updating

Grey wolves have a dominant social hierarchy that includes four levels: alpha, beta, delta, and gamma [49]. The alpha wolf is the leader, and its orders should be followed by the pack. The beta wolf is an advisor to the alpha, reinforcing the alpha’s commands to the entire pack and giving feedback to the alpha. The delta wolf has to submit to the alpha and beta but dominates the remaining pack (omega wolves). From the algorithmic point of view, at every iteration, whenever the quality of all wolves has been evaluated by Equation (29), they are sorted from the best to the worst based on their *OBJP* values. Next, the best solution, second best solution, and third best solution are considered as alpha ($X_{\alpha }$), beta ($X_{\beta }$), and delta ($X_{\delta }$), respectively, which are assumed to estimate the position of the prey. Finally, the position of the wolf *g* is updated as Equation (30).
(30)$$S_{g}\left(t+1\right)=\frac{1}{3}\left(X_{g}^{\alpha }\left(t\right)+X_{g}^{\beta }\left(t\right)+X_{g}^{\delta }\left(t\right)\right)$$
where $X_{g}^{\alpha }$, $X_{g}^{\beta }$, and $X_{g}^{\delta }$ are encircling prey factors associated with alpha, beta, and delta, respectively, which can be calculated using Equations (31)–(33).
(31)$$X_{g}^{\alpha }\left(t\right)=X_{\alpha }\left(t\right)-A_{g}^{\alpha }.\;\left|C_{g}^{\alpha }.\;X_{\alpha }\left(t\right)-S_{g}\left(t\right)\right|$$
(32)$$X_{g}^{\beta }\left(t\right)=X_{\beta }\left(t\right)-A_{g}^{\beta }.\;\left|C_{g}^{\beta }.\;X_{\beta }\left(t\right)-S_{g}\left(t\right)\right|$$
(33)$$X_{g}^{\delta }\left(t\right)=X_{\delta }\left(t\right)-A_{g}^{\delta }.\;\left|C_{g}^{\delta }.\;X_{\delta }\left(t\right)-S_{g}\left(t\right)\right|$$
where A and C are generated as A=2ar1−a and C=2r2, where r1 and r2 are vectors with uniform random components within [0, 1], and a is linearly decreased from 2 to 0 by the execution of the LSGWO algorithm. Based on the value of A, search for prey or attacking prey may be applied. A grey wolf tends to diverge from the prey if $\left|A\right|>1$ (search for prey) or converge towards the prey if $\left|A\right|<1$ (attacking prey). 

### 4.5. Local Search Improvement

At the end of each iteration, a neighbor solution $X^{*}new$ is constructed in the vicinity of the global best solution $X^{*}$ using multiple local search operators. Then, the new solution is evaluated according to Equation (29). Accordingly, the current best solution ($X^{*}$) is replaced by the new solution ($X^{*}{}_{new}$) if *OBJP**_new_ < *OBJP**. The local search process may be performed on one, two, or all six structures. In this regard, different binary and integer operators are applied based on the structure type. In the binary operator for $X_{ij}$, a hospital is randomly chosen, and its blood bank is randomly changed, i.e., a “0” is exchanged with a “1” for a hospital. It means that a previously selected blood bank is omitted and a new blood bank is replaced. To perform binary operator for $X_{ik}^{LT}$, an element of the matrix is randomly selected, and its value is complemented, i.e., the hospital k is discarded from the lateral transshipment of the hospital i, or a new hospital k is added for lateral network of the hospital i. Moreover, to perform local search for reorder points and order quantities $R_{ijl}$, $Q_{ijl}$, $R_{ikl}^{LT}$, and $Q_{ikl}^{LT}$, an element of the matrix is randomly selected and changed within the allowable range. An example for local search on the reorder point S.$R_{ijl}$ is shown in Figure 4.

## 5. Numerical Results

The proposed BPSCN model and LSGWO solution method have been successfully developed in MATLAB R2020b. All simulations have been carried out on a PC with 16 GB RAM and 2.59 GHz i7 CPU running on windows 11.

### 5.1. Simulation Settings

Settings of the controllable parameters of the LSGWO algorithm, as well as the lower/upper bounds of the decision variables, are summarized in Table 2. As GWO is a metaheuristic algorithm with auto-adjusted parameters, we should only set the algorithm’s number of iterations and population size. 

The dataset in this paper is a BPSCN with *D* = 50 donor sites, *M* = 3 blood banks, and N=12 hospitals. Every day t, each hospital i has a certain (scheduled) demand $D_{ilt}^{P}$ and an uncertain (emergency) demand $D_{ilt}^{E}$ for each age l. The aim is to design a BPSCN for 30-day time periods (one month) to minimize the total economic cost while minimizing shortage and wastage. The initial inventory of platelets with any age in each hospital was set as $I_{ilt}^{0}=30$ platelets. The disruption risks for blood banks and transportation links between blood banks and hospitals have been set as 3% and 1%, respectively. Parameters of the BPSCN model are provided in Table 3.

### 5.2. Simulation Results

To evaluate the performance of the LSGWO algorithm compared to the local search (LS) and grey wolf optimizer (GWO), each method has been implemented to solve the BPSCN model separately. Comparison of the convergence of different methods in terms of the objective function of Equation (29) versus iteration can be seen in Figure 5. Moreover, to capture the performance of the different algorithms, Table 4 statistically qualifies them in terms of the different costs and objective functions. The results demonstrate the superiority of the proposed LSGWO algorithm against both LS and GWO methods. As seen in Table 4, LSGWO has an improvement rate of 12.6% and 26.6% in the overall objective function against GWO and LS, respectively. According to Figure 5, although both GWO and LSGWO start from similar points, LSGWO has better convergence than GWO, especially during the early iterations, as it is equipped with local search operators.

### 5.3. Sensitivity Analysis

In all simulations presented in Section 5.2, the percentage of the demands for different ages of platelets have been determined as 50%, 30%, and 20%, for young, mature, and old platelets, respectively. The weights of economic, social, and environmental functions have been set as *w_EC_* = 0.5, *w_SC_* = 0.25, and *w_EN_* = 0.25, respectively. Moreover, all simulations have been performed utilizing vertical and horizontal transshipments, i.e., delivery from blood banks to hospitals (vertical coordination) and lateral transshipment between hospitals (horizontal coordination). In the following, some sensitivity analyses are provided by changes in these parameters.

To capture the effect of demands with different ages of platelets, a sensitivity analysis is shown in Table 5, wherein the shortage and wastage rates were calculated as the ratio of the average number of platelets shortage and wastage in different hospitals per day to the total received platelets in hospitals. Based on the obtained results, more demand for young platelets leads to a decrease in the wastage rate but simultaneously may increase the shortage rate.

By changing the weights of the different objective functions in Equation (13), the decision maker can design different priorities for achieving more sustainable processes according to the application specifications. The comparison of the different objective values for different weights of the objective function can be summarized in Table 6.

Finally, to capture the effect of adopting lateral transshipment between hospitals in addition to vertical transshipment (from blood banks to hospitals) on different objective functions, the model has been solved via the LSGWO algorithm with and without lateral transshipment. The comparison of the obtained results is provided in Table 7. The results demonstrate that considering the concept of lateral transshipment between hospitals significantly reduces both shortage and wastage costs by 30.1% and 18.8%, respectively. Moreover, reduction rates of 3.61% and 5.62% have been obtained in the economic cost and total objective function, respectively.

### 5.4. Comparison with Other Metaheuristics

To evaluate the performance of LSGWO against other metaheuristic algorithms, it has been compared with Genetic Algorithm (GA), Aquila Optimizer (AO) [50], and a hybrid metaheuristic algorithm based on Genetic Algorithm and Simulated Annealing (GASA) [36], to solve the BPSCN model under the same circumstances. For a fair comparison, all algorithms have been coded with the same number of objective function evaluations (NFE), considering 500 iterations and 200 solutions.

The comparison of the results obtained by different methods is provided in Table 8. The results show the superiority of the proposed LSGWO algorithm against the existing evolutionary and swarm intelligence algorithms by obtaining the minimum value for all objectives (except for the social objective of AO). Although the social objective of the AO algorithm is a little bit lower than the LSGWO algorithm, the total objective function (which is the main objective of the optimization process) of the AO algorithm is 5.8% higher than that of the LSGWO algorithm.

## 6. Conclusions

Blood platelets are highly perishable age-differentiated products that are vital for the survival of patients and for saving lives. As the human body is the sole source of blood products, blood required by hospitals should be extracted from voluntary donors, which may lead to a significant shortage (social concern) of platelets due to the limited number of donors, especially during disaster conditions such as the COVID-19 pandemic. On the other hand, the limited shelf life of blood platelets may also lead to the wastage of these products. This paper has designed a resilient–sustainable blood platelet supply chain network by considering age-differentiated demands and a vertical–horizontal transshipment strategy. The proposed model considers both planned (scheduled) and emergency (unscheduled) demands of different platelet ages. To improve the sustainability of the model, total economic cost, as well as social and environmental issues, have been taken into account as a multi-objective function. Moreover, a reactive resilient strategy has been adopted to balance shortage and waste better and make the supply chain powerful against disasters and disruption risks. The proposed model has been solved by a local search empowered grey wolf optimizer algorithm. The results have demonstrated the efficiency of the proposed method against the existing techniques.

In the proposed model, we have focused on the inventory management of blood platelets. As future work, the proposed model can be extended to other related topics, such as the network for red blood cell transfusion and the administration of coagulation plasma factors in hospitals. In the proposed model, the details of the donor sites, as well as their disruption risks, have not been under consideration. As future work, the proposed model would be modified by adding more details of the static/mobile donor sites. Typically, the process of quality inspection of platelets by the blood bank should not take more than two days to reach the hospitals. However, for the aim of simplicity, we have set it as a fixed value of two days. To make the model more consistent with the actual situation, this parameter can be set with variable values within [0, 2] or [1, 2], randomly chosen by day. In the proposed model, time periods have been considered as days, and consequently, the lead time of horizontal transshipment between hospitals (which is assumed to be done within some hours) has been ignored. As future work, the proposed model can be extended by considering more details and costs of the lateral transshipment between hospitals. Moreover, machine learning techniques can be utilized as time-series supply/demand forecasting models to increase the resiliency of the model against disruption risks such as the COVID-19 pandemic.

## Figures and Tables

**Figure 1 ijerph-20-04078-f001:**
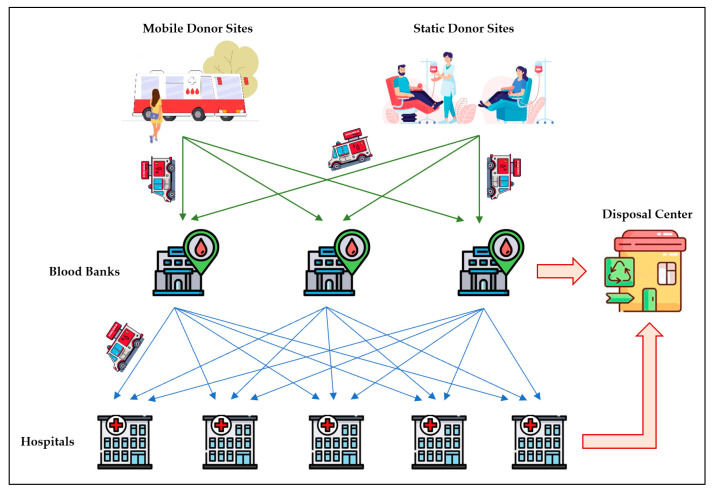
Three-echelon BPSCN model with vertical–horizontal transshipment.

**Figure 2 ijerph-20-04078-f002:**
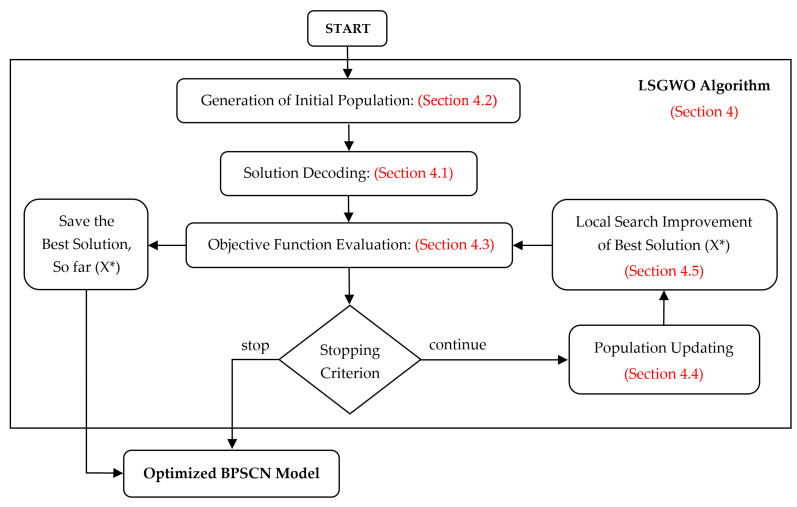
Flowchart of the LSGWO algorithm for solving the BPSCN model.

**Figure 3 ijerph-20-04078-f003:**
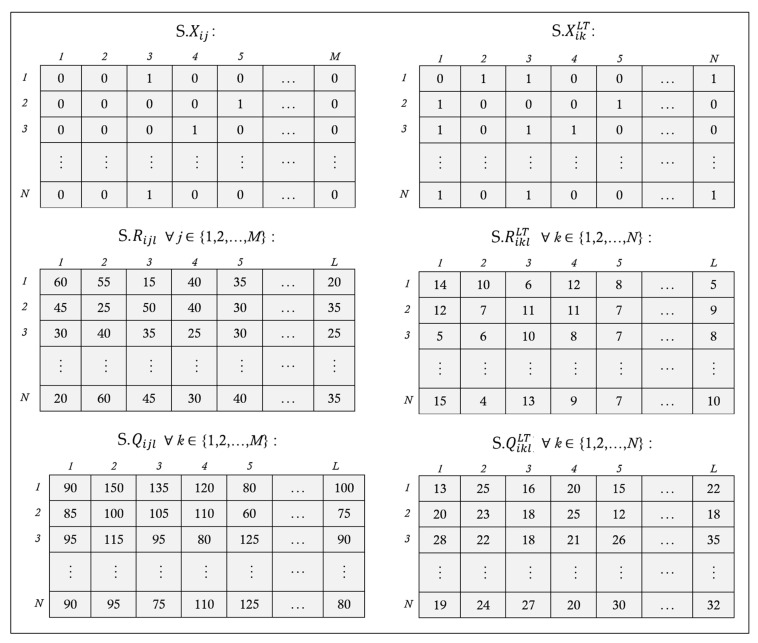
Representation of a grey wolf (i.e., a feasible solution).

**Figure 4 ijerph-20-04078-f004:**
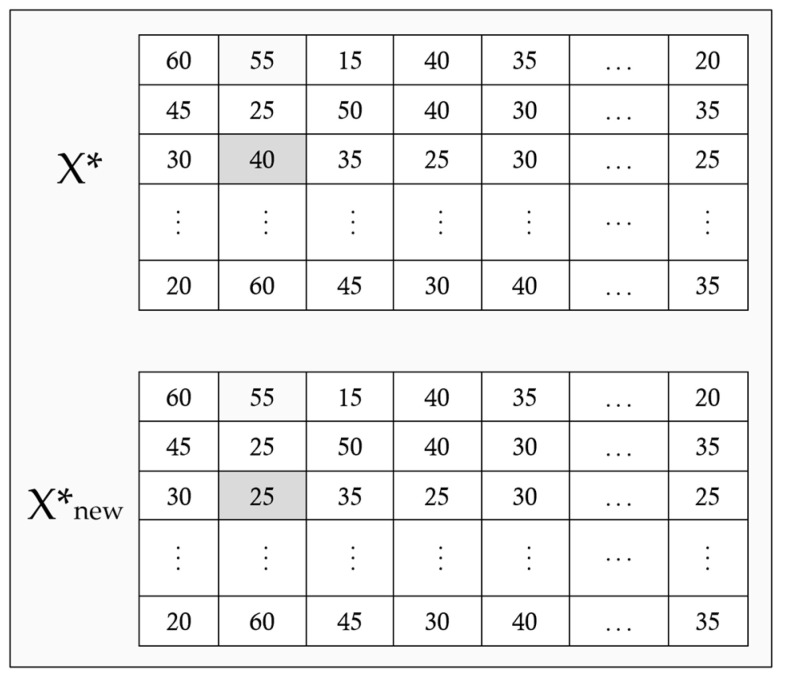
An example of integer local search for S.$R_{ijl}$.

**Figure 5 ijerph-20-04078-f005:**
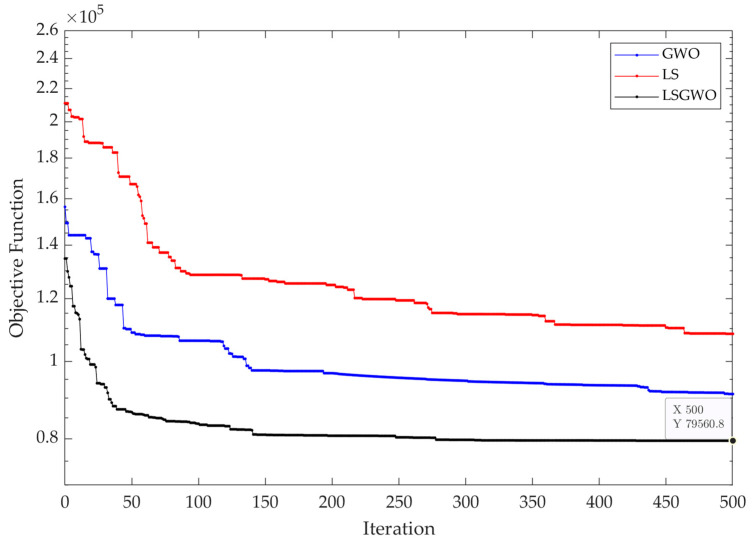
Convergence of the different methods in terms of the objective function versus iteration.

**Table 1 ijerph-20-04078-t001:** Notations.

**Indices**	**Definition**
*j*	Blood bank index, *j* = 1, 2, …, *M*
*i*, *k*	Hospital index, *i*, *k* = 1, 2, …, *N*
*d*	Donor site (supplier) index, *d* = 1, 2, …, *D*
*l*	Platelet age (young, mature, old) index, *l* = 1, 2, …, *L*
*t*	Time period (day) index, *t* = 1, 2, …, *T*
**Parameters**	**Definition**
*f_j_*	Fixed cost of ordering from blood bank *j*
*c_jl_*	Unit procurement cost by blood bank *j* for platelet age *l*
*c^LT^_kl_*	Unit interchange cost between hospitals for platelet age *l* provided by hospital *k*
*q_j_*	Percentage of platelets with acceptable quality in blood bank *j*
*h_il_*	Unit holding cost for platelet age *l* in hospital *i*
*p_ij_*	Fixed transportation cost per order to hospital *i* from blood bank *j*
*r_ij_*	Variable mileage cost per order to hospital *i* from blood bank *j*
*d_ij_*	Distance between hospital *i* and blood bank *j*
*p^LT^_ik_*	Fixed transportation cost per lateral transshipment to hospital *i* from hospital *k*
*r^LT^_ik_*	Variable mileage cost per lateral transshipment to hospital *i* from hospital *k*
*d^LT^_ik_*	Distance between hospital *i* and hospital *k*
*L_dj_*	Lead time between donor site *d* and blood bank *j*
*L_ij_*	Lead time between hospital *i* and blood bank *j*
*L^LT^_ik_*	Lead time between hospital *i* and hospital *k*
*s_il_*	Unit shortage cost for platelets with age *l* in hospital *i*
*w_i_*	Unit wastage cost in hospital *i*
*w_j_*	Unit wastage cost in blood bank *j*
*D^P^_ilt_*	Planned (scheduled) demand for platelets with age *l* in hospital *i* on day *t*
*D^E^_ilt_*	Emergency (unscheduled) demand for platelets with age *l* in hospital *i* on day *t*
*D_ilt_*	Total demand for platelets with age *l* in hospital *i* on day *t*, *D_ilt_ = D^P^_ilt_ + D^E^_ilt_*.
*I* ^0^ * _ilt_ *	Inventory of platelet age *l* in hospital *i* at the beginning (morning) of day *t*
*I_ilt_*	Inventory of platelets with age *l* in hospital *i* at the end (night) of day *t*
*S_jdt_*	Quantity of blood platelets donated from donor site *d* to blood bank *j* at time *t*
*TPT_j_*	Testing and producing time in blood bank *j*
*IB_jlt_*	Inventory of platelets age *l* in blood bank *j* at time *t*
*Z_jd_*	Binary parameter and equal to 1 if blood bank *j* is sourced by donor site *d;* 0, otherwise
*E_ilt_*	Binary parameter and equal to 1 if total demand *D_ilt_* is more than the inventory *I*^0^*_ilt_*; 0, otherwise
**Decision Variables**	**Definition**
*X_ij_*	Binary variable and equal to 1 if hospital *i* is supplied by blood bank *j*; 0, otherwise
*X^LT^_ik_*	Binary variable and equal to 1 if hospital *i* is supplied by hospital *k*; 0, otherwise
*Y_ijlt_*	Binary variable and equal to 1 if hospital *i* is sourced by blood bank *j* for platelets with age *l* at time *t*; 0, otherwise
*Y^LT^_iklt_*	Binary variable and equal to 1 if hospital *i* is sourced by hospital *k* for platelets with age *l* at time *t*; 0, otherwise
*R_ijl_*	Reorder point of hospital *i* for platelets with age *l* from blood bank *j*
*Q_ijl_*	Order quantity of hospital *i* for platelets with age *l* from blood bank *j*
*R^LT^_ikl_*	Reorder point of hospital *i* for platelets with age *l* from hospital *k*
*Q^LT^_ikl_*	Order quantity of hospital *i* for platelets with age *l* from hospital *k*

**Table 2 ijerph-20-04078-t002:** Parameters of the LSGWO algorithm.

Parameter	Value/Description
Maximum number of iterations (MaxIter)	500
Population size of grey wolves (PopSize)	200
Multiple local search operators	Binary Swap/Exchange, Integer Swap
Weight of economic objective (*w_EC_*)	0.5
Weight of social objective (*w_SC_*)	0.25
Weight of environmental objective (*w_EN_*)	0.25
Lower and upper bounds of reorder point from blood banks	[5, 30]
Lower and upper bounds of order quantity from blood banks	[20, 100]
Lower and upper bounds of reorder point for lateral transshipment	[2, 10]
Lower and upper bounds of order quantity for lateral transshipment	[5, 20]

**Table 3 ijerph-20-04078-t003:** Parameters of the BPSCN model.

**Parameter**	**Value**	**Parameter**	**Value**
** *f_j_* **	20	*S_jdt_*	(20–60)
*q_j_* (∀j)	0.85	*D^P^_ilt_*	(50–100)
*s_il_* (∀l)	0.5	*D^E^_ilt_*	(0–30)
*w_i_*	0.3	*TPT_j_*	2
*p_ij_*	3	*L_dj_*	0
*r_ij_*	1.5	*L_ij_*	1
*p^LT^_ik_*	2	*L^LT^_ik_*	0
*r^LT^_ik_*	2	*I*^0^*_ilt_* (*t* = 0)	30
**Parameter**	**Value for Young (*l* = 1)**	**Value for Mature (*l* = 2)**	**Value for Old (*l* = 3)**
*c_jl_*	2.5	2	1.5
*c^LT^_kl_*	0.3	0.25	0.15
*h_il_*	0.1	0.075	0.05
Demand %	50	30	20

**Table 4 ijerph-20-04078-t004:** Detailed results of the objective function for different methods (in 1000$).

Objective Function	GWO	LS	LSGWO	Improvement of LSGWO	
				Against GWO	Against LS
Cost of ordering from Blood banks	120	141.4	102.8	14.3%	27.3%
Cost of lateral Transshipment	8.51	8.99	9.33	−9.64%	−3.78%
Cost of transportation	22.6	29.3	22.3	1.33%	23.9%
Cost of inventory holding	17.8	18.6	14.4	19.1%	22.6%
Economic objective (total cost)	168.91	198.29	148.83	11.9%	24.9%
Social objective (shortage)	14.7	19.1	11.4	22.4%	40.3%
Environmental objective (wastage)	11.6	17.9	9.17	20.9%	48.8%
Overall objective function	91.03	108.4	79.56	12.6%	26.6%

**Table 5 ijerph-20-04078-t005:** Effect of demands with different ages of platelets on shortage and wastage rates.

Percentage of Demand for			Shortage Rate	Wastage Rate
Young	Mature	Old		
50%	30%	20%	6.87%	10.31%
40%	30%	30%	6.53%	10.87%
30%	20%	50%	5.2%	12.3%
20%	30%	50%	5.48%	11.92%
60%	30%	10%	7.46%	9.41%
70%	20%	10%	8.12%	8.8%

**Table 6 ijerph-20-04078-t006:** Effect of changes in the weights of the three objective functions on economic, social, and environmental costs (in 1000$).

*w_EC_*	*w_SC_*	*w_EN_*	Economic Cost	Social Cost	Environmental Cost
0.5	0.25	0.25	148.83	11.4	9.17
1	0	0	113.7	22.8	13.76
0	1	0	162.49	5.31	16.11
0	0	1	145.54	23.7	4.35
0.25	0.5	0.25	157.21	8.1	9.82
0.25	0.25	0.5	153.8	12.6	7.86

**Table 7 ijerph-20-04078-t007:** Comparison of objective functions (in 1000$) with and without lateral transshipment.

**Cost**	**Without LT**	**With LT**	**% Improvement of LT**
Cost of ordering from Blood banks	118.7	102.8	13.4%
Cost of lateral Transshipment	0	9.33	N/A
Cost of transportation	18.4	22.3	−21.2%
Cost of inventory holding	17.3	14.4	16.8%
Economic objective (total cost)	154.4	148.83	3.61%
Social objective (shortage)	16.3	11.4	30.1%
Environmental objective (wastage)	11.3	9.17	18.8%
Overall objective function	84.3	79.56	5.62%

**Cost**

**Table 8 ijerph-20-04078-t008:** Comparison of the different objectives for different algorithms (in 1000$).

Objective Function	GA	AO [50]	GASA [36]	LSGWO
Economic objective (total cost)	163.45	156.9	153.61	148.83
Social objective (shortage)	13.6	11.1	12.8	11.4
Environmental objective (wastage)	12.5	12.8	10.3	9.17
Overall objective function	88.25	84.43	82.58	79.56

## Data Availability

The data used in the study are available from the authors and can be shared upon reasonable requests.

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
