# Peer review of "Designing a Resilient–Sustainable Supply Chain Network of Age-Differentiated Blood Platelets Using Vertical–Horizontal Transshipment and Grey Wolf Optimizer"

_ijerph, 2023, doi:10.3390/ijerph20054078_

Round 1
Reviewer 1 Report
I enjoyed reading this manuscript. It is well organized and written down. But please put some more efforts on English. Please make sure all of references have the same format.
Reviewer 2 Report
Shokouhifar et al submit an interesting paper regarding the Resilient-Sustainable Supply Chain Network of 2 Age-Differentiated Blood Platelets. I have several edits and comments that need to be addressed before further assessing the manuscript.
General comments
1. The english used in the manuscript could be edited for further improvement.
2. The methods described in this work are clearly presented.
3. Introduction - the authors are advised to remove the bullet points from that section and describe their points in a concise manner. The introduction is adequate for the reader to understand the research question.
4. In figure 6, I am not sure if I fully understand the message that the authors would like to pass.
5. Given the interesting approach of this work, I was wondering if the authors could discuss potential applications of this model in other related topics such as the network for red blood cell transfusion and the administration of coagulation plasma factors in hospitals.
Reviewer 3 Report
Shokouhifar and Goli focused on blood platelet supply under disaster conditions. A so-called BPSCN (blood platelet supply chain network) was constructed to manage the platelet supplement in an efficient and cost-effective manner.
In general, I think the authors did quite some work, and the results are convincing and interesting. The hypothesis and implementation process of the author's model are detailed and consolidated. Although some details need further explanation, I have some questions related to the methods and results. I have some major criticisms. Here are some questions and comments that the authors should think about as they make changes to their work.
1. The TPTj was set as a constant and represents 2 days. This seems to be inconsistent with the actual situation. The quality inspection of platelets by the blood bank should be started after collecting a certain amount of raw materials, and this process should not take more than 2 days to reach the hospital. Although the author also stated in the discussion that more details need to be added, the quality inspection time should not have a fixed value. If the author can set the quality inspection time as a range and assign values randomly by day, it may be more consistent with the actual situation.
2. The lead time for horizontal communication between hospitals has been ignored. This assumption is very unreasonable. Platelet exchange between hospitals is an important parameter in the model and an important factor in reducing waste. The author should not assume that this time does not exist because the physical distance between hospitals is one of the important factors that affect the exchange time, and the distance is also a parameter of the exchange cost. If there is no lead time, it means that the hospital needs to respond quickly. This is obviously not in line with the actual situation of the hospital. In addition, hospitals must establish a set of information systems to exchange information among themselves to clarify important information such as inventory and estimated usage, which are factors that increase costs. If the exchange lead time does not exist, the above cost increase factors will be more difficult to control.
3. The qj was set at 0.85. Is this an empirical value, or is it supported by survey data?
4. The Sil was set at 0.5. If different Sis of different ages were the same, why define the Sil?
Round 2
Reviewer 3 Report
The authors responded to all of my concerns carefully. I have no further questions.